# 3D designed and printed chemical generators for on demand reagent synthesis

Sergey S. Zalesskiy [1], Philip J. Kitson[1], Przemyslaw Frei[1], Andrius Bubliauskas[1] & Leroy Cronin [1]*

Modern science has developed well-defined and versatile sets of chemicals to perform many specific tasks, yet the diversity of these reagents is so large that it can be impractical for any one lab to stock everything they might need. At the same time, isssues of stability or limited supply mean these chemicals can be very expensive to purchase from specialist retailers. Here, we address this problem by developing a cartridge -oriented approach to reactionware-based chemical generators which can easily and reliably produce specific reagents from low-cost precursors, requiring minimal expertise and time to operate, potentially in low infra-structure environments. We developed these chemical generators for four specific targets; transition metal catalyst precursor tris(dibenzylideneacetone)dipalladium(0) [$Pd_2(dba)_3$], oxidising agent Dess-Martin periodinane (DMP), protein photolinking reagent succinimidyl 4,4'-azipentanoate (NHS-diazirine), and the polyoxometalate cluster {$P_8W_{48}$}. The cartridge synthesis of these materials provides high-quality target compounds in good yields which are suitable for subsequent utilization.

[1] School of Chemistry, The University of Glasgow, Glasgow G12 8QQ, UK. *email: lee.cronin@glasgow.ac.uk

The power of modern chemistry, biology and material science is shown by a range of capabilities such as the ability to design a single-step assembly of complex molecules[1], targeted gene modifications[2], and property-driven materials discovery and manufacturing[3]. However in order to fulfil these complicated tasks a set of no less complex tools is always needed. In modern synthetic chemistry one routinely uses dedicated catalyst-ligand systems to direct the transformation towards a single desired pathway[4]. Furthermore, molecular biologists benefit from a range of cleaving/cross-linking agents capable of selective action at a defined site[5]. Unfortunately while all these tools open a whole world of opportunities, their preparation is often tedious[6] and time-consuming[7], requiring more time and effort than using them later. In order to run a process faster or in a more selective and reliable manner, one often needs to spend equivalent or larger amount of time preparing catalysts, ligands, biomarkers or selective binding agents. Fortunately, key industry players quickly recognized the needs of modern research and nowadays they provide a broad range of kits used in chemistry[8] and/or molecular biology[9]. However, buying reagent kits has its own limitations. First, despite the progress in modern computational chemistry, we are still far from confident prediction of reaction outcomes/synthesis planning, so it is often impossible to identify the optimal reagent of choice without trying it. This in turn means that every single lab must stock not only one of those modern highly complex (and thus expensive) reagents but rather a set of them which linearly increases the costs. Besides that, many of the reagents used in the synthetic chemistry are not stable upon storage, so it makes little sense for any lab not dealing with that exact type of process to constantly keep a wide library of catalysts/ligands, etc. To minimize cost and lead times researchers often choose to prepare the reagents themselves instead of buying them. This approach leverages these issues at a cost of the time of the researcher, which is indisputably the most expensive resource in the modern synthetic chemistry laboratories[10].

A common way to effectively generate unstable reagents has been first introduced with micro/millifluidic-based devices called chemical generators[11]. Using simple flow reactors researchers demonstrated synthesis and in situ utilization of reagents that would be unstable under normal conditions such as singlet oxygen[11], diazomethane[12], CF3-radicals[13] and others[14–18]. However, this concept suffers from all classical problems of flow protocols, such as occasional flow path blockage by solids, and, most important, need of highly skilled personnel to operate the devices.

Recently we demonstrated an approach to digitize multi-step organic syntheses for further running with minimal human intervention[19]. The key finding was the workflow to create a digital synthesis blueprint which, once validated, could have been easily reproduced an infinite number of times in a highly automated manner. Such blueprints would in future form an extensive on-line library of published synthetic protocols available to researchers from all over the world. Translating organic chemistry into a digital form also brings in benefits of ease of storage/licensing/distributing/version control. Here we demonstrate how it is possible to expand this approach to alleviate the problems summarized above, dramatically saving the time and money of expert researchers while providing access to commonly used organic reagents with minimal effort.

## Results

### Choice of synthetic targets
The two main challenges of using modern synthetic chemistry/molecular biology toolkit are stock supply (with all associated costs and delivery expenses) and storage (including stability and storage organization issues). While the first issue is efficiently mitigated by in-house synthesis, the second problem remains, and often requires on-demand repeats of small-scale reagent preparation if the use of freshly prepared reagents is crucial[20]. The digitization protocol not only solves the storage problem (as one can stock and store cartridges loaded with stable precursors generating reagents on demand), but also significantly decreases the cost as the commercial prices for the reagents depend not only on the cost of the substrates/difficulty of synthesis but are also influenced by logistical and market forces. The material cost difference (i.e. excluding human resource/infrastructure) between synthesizing and buying a reagent from a company can be clearly seen in Fig. 1. The compounds used as examples were chosen on the basis of their wide use and applicability in different areas of chemistry. Compound A, succinimidyl 4,4′-azipentanoate, commonly known as NHS-diazirine, is a next-generation cross-linking agent widely used in molecular biology for cross-linking proteins[21–23]. Bifunctional linking (amine-reactive NHS site complemented with UV-activated diazirine cleavage) provides more versatility compared to traditional azide-based linking agents[24]. However, the wide popularity of this class of linking agents among biologists is somewhat hampered by the exceptionally high price of purchasing the material commercially (Fig. 1a). As one can see, for in-house preparation the estimated costs are around $4,000 USD per mole of material synthesized, while the price from the supplier is more than one million USD per mole. Inevitably, this calculation is approximate and does not account for other factors such as manpower, energy, depreciation expense, waste treatment and many others, which will vary depending on the specific situations the syntheses are performed in. We present this comparison here simply as a minimal metric to enable rough comparison between the differing approaches. The other two examples belong to transition metal catalysis and classical organic synthesis, respectively (Fig. 2). Tris(dibenzylideneacetone)dipalladium(0) (Pd2dba3 always forms a solvate in crystal form. One of the most common variations which was used in the current study is Pd2dba3·CHCl3 adduct, further referred to as just "Pd2dba3".) is one of the most common Pd(0) precursors for catalytic reactions[25]. Due to ease of use and relative stability this compound is widely utilized for cross-coupling reactions[26], C–H activation processes[27,28], mechanistic investigations[28,29] and many other

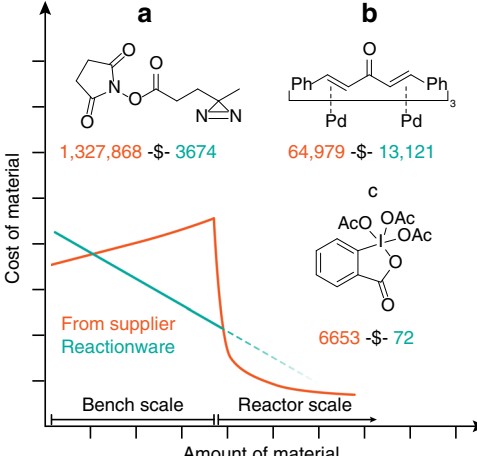

**Fig. 1** Why synthetic chemists sacrifice their precious time doing routine chemistry. Comparison of raw material price per mole for three examples (**a** succinimidyl 4,4′-azipentanoate, **b** tris(dibenzylideneacetone) dipalladium(0) and **c** the Dess-Martin periodinane) of self-made (teal) and commercially sourced (orange) reagents. Comparison necessarily excludes contingent factors such as manpower, energy, depreciation expense and waste treatment among others.

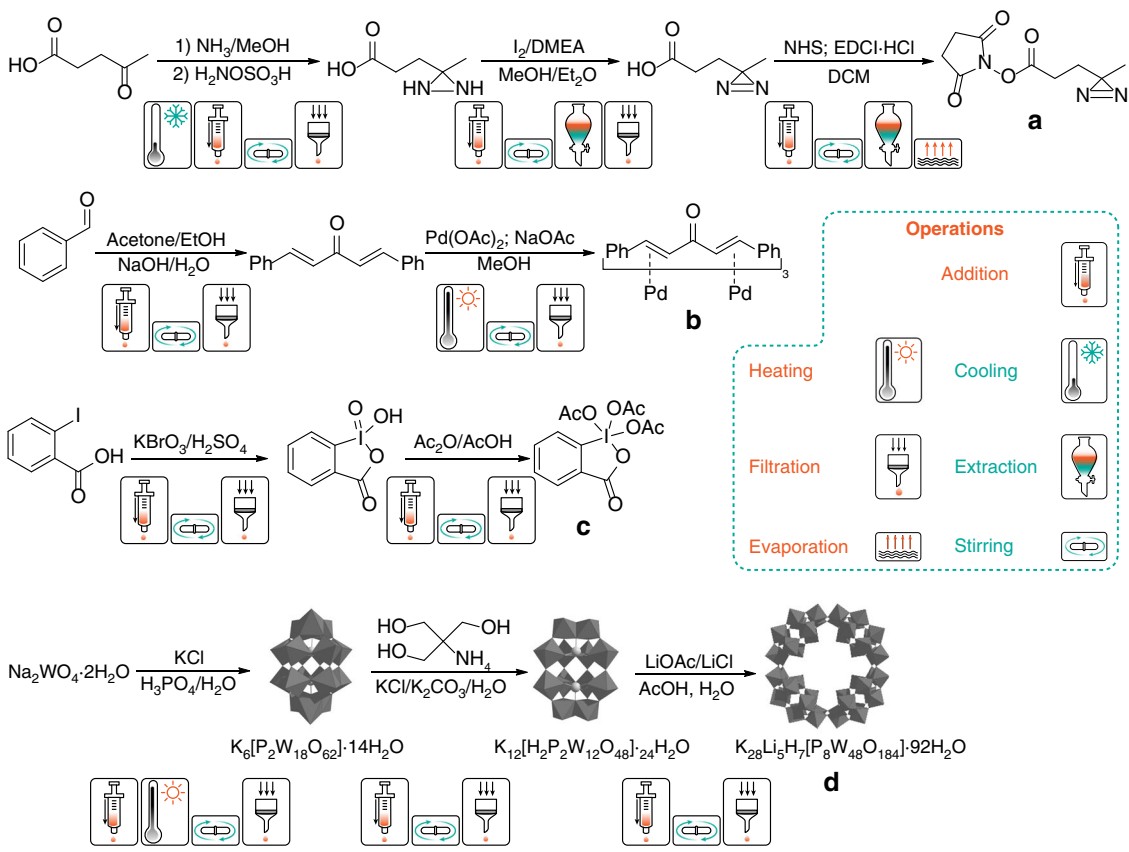

**Fig. 2** Targets chosen to demonstrate utility and versatility of reactionware approach. The synthetic routes chosen to be implemented for each of the selected target compounds, **a** succinimidyl 4,4′-azipentanoate, **b** tris(dibenzylideneacetone)dipalladium(0), **c** the Dess-Martin periodinane and **d** the {P₈W₄₈} POM cluster, along with the operations necessary for achieving the syntheses.

types of reactions[30]. The most common alternative, Pd(PPh₃)₄ is unstable to air[31] mandating the use of glovebox or Schlenk techniques, although it is often still the compound of choice for accurate mechanistic studies due to easier release of phosphine ligands compared to dba[32]. The cost estimation shows that for Pd₂dba₃ the difference, although being not as dramatic as for the NHS-diazirine, is still almost five-fold. However, there is another more important reason to choose this compound as a relevant example. Recently it has been shown that Pd₂dba₃ decomposes upon storage (which is accompanied by release of free Pd(0) nanoparticles) independent of storage conditions—inert atmosphere, air, room temperature, freezer, etc., so that 6 months after purchase, the reagent might already be of unacceptable purity[33].

The third compound listed is a well-known oxidation agent, 1,1,1-triacetoxy-1,1-dihydro-1,2-benziodoxol-3(1H)-one (Dess–Martin periodinane (DMP)) was discovered around 1983 by Daniel Dess and James Martin[34]. It demonstrates high alcohol affinity and is the reagent of choice for selective alcohol group oxidation to corresponding aldehydes or ketones in the presence of other oxidation-sensitive groups[35,36]. Compared to its predecessor, 2-iodoxybenzoic acid (IBX), DMP is more stable and typically shows higher reactivity and much better solubility in organic solvents due to the presence of acetoxy groups[37]. It has been demonstrated that compared to other common oxidants based on chromium salts or sulfur ylides, DMP often requires milder conditions thus enabling conversion of challenging and valuable substrates[38,39]. DMP is easily prepared from IBX (Fig. 2), so the use of freshly prepared DMP would often be a good choice.

The price difference in this case is about 100-fold; thus, in-house preparation would be a tempting option. However, even though this two-step synthesis is rather simple and

straightforward, it would still require around 1.5–2 days of a trained researcher's time and attention considering all involved activities (weighing the reagents, preparing the glassware, isolating the product, cleaning the glassware, checking purity). Translating all these syntheses into reactionware provides substantial time saving as no pre-/post-synthesis preparations are needed and the cartridges themselves are deemed to be single-use devices, thus requiring no cleaning as well.

Another interesting question would be whether the process stability inherent to reactionware would be able to effectively mitigate reproducibility issues often encountered in modern chemistry[40]. This is a particular problem in the area of complex inorganic compounds synthesis, such as POMs as a multitude of parameters need to be precisely maintained to yield the desired product[41]. To demonstrate the benefits of the synthesis in reactionware we decided to choose {P₈W₄₈} POM cluster as a challenging target. This tungsten polyanion exhibits an internal cavity which has been utilized for host–guest chemistry. Various transition metal-substituted {P₈W₄₈} derivatives have been reported such as the well explored copper {Cu₂₀XP₈W₄₈}, where $x = (N_3)_6$, Cl, Br, I[42,43]. Cu derivatives have been known to form supramolecular blackberry structures in aqueous solutions, useful in catalysis. Iron {Fe₁₆P₈W₄₈} also showed potential for use in catalysis[44]. Cobalt, vanadium and nickel atom centres were incorporated into the central cavity and displayed interesting electrochemical and magnetic properties[45]. The outer surface of {P₈W₄₈} can also be functionalized with transition metals. Cobalt, nickel and manganese derivatives have been shown to form porous frameworks. Furthermore, cobalt derivatives were shown to form flexible inorganic materials[46,47].

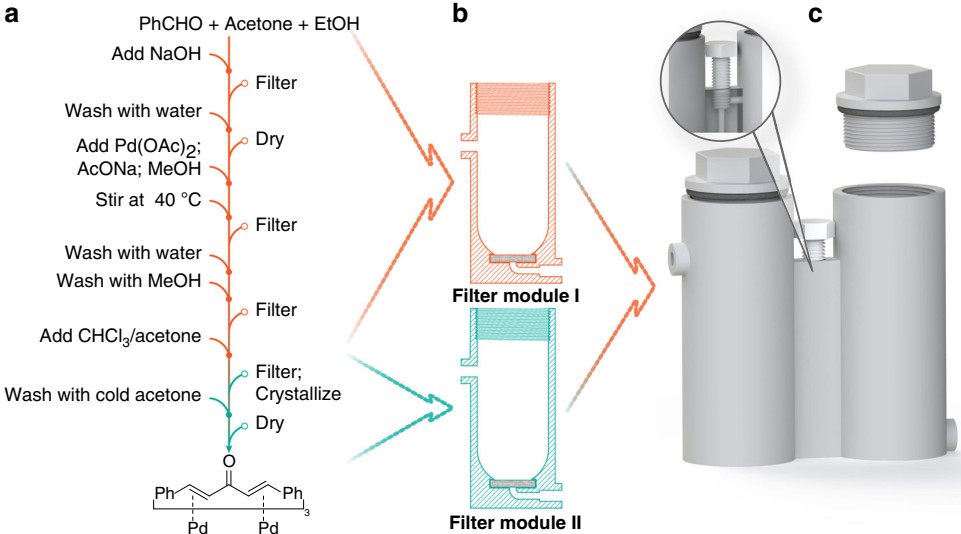

**Fig. 3** Digitization of synthesis of $Pd_2dba_3$. **a** Identification of key processes; **b** grouping processes and mapping them to reaction modules (in this case, two identical filter modules); **c** final monolithic cartridge design with two modules digitally stitched together. The screw valve operation principle is shown in the cut-out.

**Synthesis translation**. In order to translate the synthetic processes to the reactionware approach we follow the protocol described in our previous study. After testing telescoped synthetic procedures in glassware, they are converted into a sequence of elementary steps representing individual acts of moving the fluid (Fig. 3a). After analysis, these steps are grouped by module type needed to run them (Fig. 3b). Individual modules are then grouped together to form a single cartridge device to run the whole process from beginning to end (Fig. 3c). At that stage we have fully formed three-dimensional (3D) models of the cartridge which can be prototyped using a range of techniques. For optimization purposes we use fused deposition modelling (FDM) 3D printing as a sensible combination of speed, cost and design restrictions. Polypropylene is used as a material for manufacturing the cartridges due to its unique combination of high chemical resistance, relative temperature stability, low price and moderate viscosity in the molten state. The latter property is of particular importance for FDM 3D printing, as many fluorinated polymers (such as PTFE and FEP), while demonstrating excellent chemical resistance, have too high viscosity in the molten state to be used as the materials for FDM 3D printing. The cartridge design has several new features compared to those used in previous work[19]. In particular, each individual chamber is equipped with a screw cap, which provides for ease of access to the module interior during the synthesis. Besides allowing us to check and troubleshoot individual steps in the monolithic cartridge, this also significantly decreases printing time. Standard plumbing BSPP ½″ or ¾″ polypropylene caps fitted with a rubber gasket are used to provide an adequate seal on pre-tapped reaction chambers (Fig. 3c). If extra fluidic connections have to be attached to the top, a cap can be equipped with standard Luer-lock ports. Another issue that frequently impeded the operations in the older generation of reactionware was uncontrolled liquid transfer between the modules. In certain circumstances, it was found that the pressure in one of the modules could increase rapidly, due to exothermic reactions or the evolution of gas in the course of a reaction. If the openings are not wide enough to relieve the pressure immediately, it would then drive the liquid from the bottom of one module to the top of the adjacent one through the siphon tube before the desired process was complete.

In order to circumvent the build-up of pressure differentials between the modules, we introduced screw valves between individual units. These valves do not require any pre-modifications to the design files; however, they make the manual operations much more reliable. In order to prepare the valves a blind 6.8 mm diameter hole is drilled from the top surface of the module junctions to the upper bend of the siphon tube. The hole is then tapped with an M8x1.25 thread and the nylon screws are screwed in. Such screws are quite common hardware components and can be found from a range of suppliers. The bottom of the screw presses into the siphon tube preventing liquid being pushed through (Fig. 3c, cut-out).

**Cartridge design and synthesis of $Pd_2dba_3$**. The cartridge for the synthesis of $Pd_2dba_3$ consists of two filter modules, as shown above (Fig. 3). A filter module is a simple cylindrical reaction chamber with a fritted glass filter installed in the bottom. A pause is introduced in the printing process to install such non-printed components (glass frit in this case). The cartridge walls are designed to overhang on top of the filter to provide an effective seal. Both filter modules are equipped with screw caps, the first one carrying a thermometer to enable temperature control and the second one a single female Luer-lock port. The process is started by running a double crotonic condensation of acetone with benzaldehyde to yield dibenzylideneacetone. The product of the first step is then filtered and dried in the first module after which the product is re-dissolved in methanol, and followed by addition of palladium (II) acetate and sodium acetate. After subsequent heating, the reaction mixture is filtered and the solid residue washed to yield $Pd_2dba_3$ in module 1. The crude product is re-dissolved in chloroform and transferred through the filter and siphon tube to module 2 where it is left to crystallize. All potential solid impurities are left on the filter. The drawings of the cartridge and details of the reaction protocol can be found in Supplementary Methods. Full reaction sequence took approximately 84 h and yielded product with 95% yield (91% purity).

**Cartridge design and synthesis of diazirine-NHS**. The synthesis of the NHS-diazirine adduct is by far the most complicated

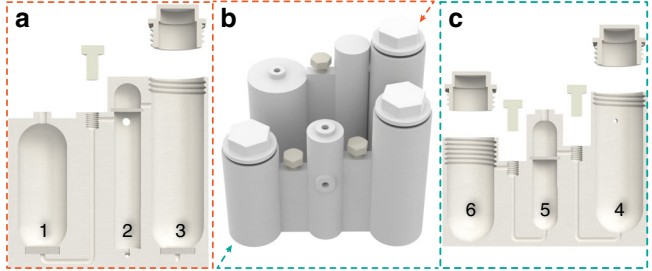

**Fig. 4** Cartridge design for the synthesis of NHS-diazirine. Complete cartridge (**b**) and cross-sections along the top (**a**) and bottom (**c**) rows of modules. Numbers correspond to individual modules (see text).

reaction sequence implemented in reactionware. Despite apparently straightforward protocol (Fig. 2a), it requires a lot of liquid manipulation steps resulting in a cartridge consisting of six different modules (Fig. 4).

The synthesis is started by reacting levulinic acid with ammonia in methanol in module 1 (Fig. 4a) followed by in situ addition of hydroxylaminesulfonic acid. After that the reaction mixture is filtered off to the bottom of module 2, vacuum-dried and re-dissolved in methanol. Dimethylamine and iodine are added to oxidize the N–N bond in the diaziridine ring (Fig. 2a, step 2). After completion the reaction is quenched with KI/ascorbic acid/HCl solutions, which are introduced sequentially through the upper side port of module 2. Addition of the diethyl ether through bottom port of module 2 provides extraction of the product into organic phase and pushes it through the hydrophobic membrane installed at the top of the module 2 to module 3, which is pre-loaded with $MgSO_4$ before reaction start, through a short coupling. After drying, external pressure is used to filter product solution off and transfer it to module 4 via a siphon pipe. In module 4 the solvent is swapped for DCM and an EDCI-mediated coupling with NHS takes place. The reaction mixture is quenched with water, transferred to module 5 and pushed through a hydrophobic membrane to the lower compartment. The lower compartment of module 5 is preloaded with dry molecular sieves. After removing traces of water, the solution is transferred to the last module (6), the solvent is removed under reduced pressure, the crude product is redissolved in the $MeOH/Et_2O$ mixture and left to crystallize. The protocol yields desired NHS–diazirine product in 20% yield with 90% purity. The drawings of the cartridge and details of the reaction protocol can be found in Supplementary Methods.

**Cartridge design and synthesis of Dess-Martin periodinane**. Dess-Martin periodinane is synthesized following a classical protocol[48] starting from 2-iodobenzoic acid. This route is often followed in the laboratories to prepare fresh DMP; however, it involves IBX as an intermediate product which has to be handled with care due to its explosive nature[49]. Containing the full synthetic procedure in a cartridge significantly lowers the risks, as no explicit manipulations with unstable compounds are performed. The reaction cartridge for the synthesis of DMP represents a single filter module equipped with an in-printed drying tube filled with $CaCl_2$ at the print time. After the first step, the 2-iodoxybenzoic acid formed is washed on the filter and left to dry inside the cartridge. $Ac_2O/AcOH$ mixture is added to the same vessel and the reaction mixture is stirred at 80 °C. After the reaction completion it is filtered off and the solid residue is washed with $Et_2O$ to give DMP with 69% yield and 81% purity.

**Cartridge design and synthesis of $W_{48}$**. As summarized above, $\{P_8W_{48}\}$ POM is a challenging target due to the need of very fine

control of reaction pH and high dilution ratios necessary for the synthesis to succeed. Due to the large physical size of the complete cartridge (approximately $130 \times 180$ mm footprint and total weight around 550 g) it is printed as two separate physical pieces which are connected together with a short section of tubing equipped with Luer-lock connectors on both ends. The drawings of both cartridges can be found in Supplementary Figures. For the first step, module 1 is charged with $Na_2WO_4$ followed by addition of $H_3PO_4$. After equilibrating the pH, the reaction is microwaved for approximately 1 h. As polypropylene has negligible absorbance in the microwave region[50], it makes possible to run microwave heating without any risk of melting the vessel. Compared to the glassware protocol (24 h reflux), this is a substantial time saving. After the reaction cools down, it is transferred to the next module loaded with $NH_4Cl$ by applying external pressure to module 1. After formation of precipitate it is transferred to module 3 and filtered. Here an extra precaution is taken by running the reaction in a separate module before filtering, so that no precipitate would be able form underneath the filter from the liquid that has passed through over the course of the reaction. KCl solution is added to module 3 and the resulting precipitate of $\{P_2W_{18}\}$ is filtered off and dried by passing an air stream through the cartridge. To run the next step, $\{P_2W_{18}\}$ is re-dissolved in water and siphoned from module 3 (cartridge 1) to module 4 (cartridge 2) to which a solution of TRIS is added. The resulting solution is then transferred to module 5 which has been preloaded with KCl and the mixture is stirred and subsequently transferred to module 6, where a solution of $K_2CO_3$ is added. A suspension of $\{P_2W_{12}\}$ forms and is filtered off and dried. Finally, $\{P_2W_{12}\}$ is re-dissolved in a buffer solution in the same module, the solution is filtered through the frit in the module bottom and left to crystallize yielding 29% of the target $\{P_8W_{48}\}$ cluster.

**Comparison of cartridge with traditional synthesis**. The four target compounds described were synthesized in comparable yields and purities to traditional glassware syntheses. In order to demonstrate the utility of the generated chemical products a series of validation experiments were run in which the reactivity of the reactionware synthesized targets were compared with commercially obtained samples of the same products (except in the case of the $\{P_8W_{48}\}$ cluster, which is not available from commercial suppliers—in this case a literature procedure using the cluster as a starting material was repeated to validate its reactivity). In each case the behaviour of the reactionware produced material was comparable to the commercial samples, showing their suitability as replacements for the commercial products (see Methods below). The use of the cartridge architecture along with the clear, precise step-by-step instructions (detailed in the Supplementary Methods) for the synthesis of each target dramatically reduces the expertise necessary for successful synthesis of each compound. This results in time-saving efficiencies in both quantity and quality of necessary interactions with the synthesis. That means that the cartridges require less complex and time-consuming interventions than traditional syntheses. This, along with smaller cartridge footprint compared to glassware setups, makes it possible for a single researcher to run more syntheses in parallel, or for the syntheses to be accomplished by researchers with lower practical skill levels. The exact amount of time saved in terms of being able to run more reactions in parallel and simplified operation of the cartridges is of course very subjective and would vary from one researcher to another. However, we did an analysis and this is summarized in Table 1, and is based on our extensive experience using cartridge devices. One can see that although the individual time saving is modest, the ability to run multiple reactions in parallel drives

**Table 1 Time benefits of reactionware approach compared to traditional synthesis.**

| Researcher's time consumed (hours) | | Time boost | Compound | Parallel runs per researcher | | Parallelization boost |
|---|---|---|---|---|---|---|
| Manual vs cartridge | | | | Manual vs cartridge | | |
| 12 | 10 | +17% | $Pd_2dba_3$ | 2 | 5 | ×2.5 |
| 16 | 13 | +19% | DMP | 1 | 7 | ×7 |
| 24 | 20 | +17% | NHS-diazirine | 2 | 4 | ×2 |
| 24 | 18 | +25% | $\{P_8W_{48}\}$ | 2 | 3 | ×1.5 |

reactionware far ahead of conventional synthesis. The former can be easily explained by the fact that for majority of organic reaction the actual reaction time is unchanged, and the difference arises from the lack of glassware preparation/assembly/cleaning procedures. However, this is not the case for the synthesis of $\{P_8W_{48}\}$, where the efficiency of microwave heating in the reactionware yields significant time saving, as described above. The parallelization boost is much more pronounced though. The reactionware cartridges, being small monolithic leak-proof devices, require much less attention during operating and consume much less bench/fumehood space, thus enabling researcher to run more parallel reactions if needed. One may argue, that for traditional bench-top chemistry the upscaling is the way to go and a chemist would just swap 50 mL flask for a 1 L flask instead of running 20 flasks in parallel; however, a crucial point here is that different cartridges (meaning different multi-step chemical processes) can be similarly run in parallel, which is plainly impossible with the traditional approach.

Another issue that often comes along the topic of plastic cartridge development and acceptance is the environmental impact. Several points need to be adressed here. First, it is obviously impractical to suggest that cartridges should completely replace the modern reusable glassware. However, in certain circumstances, such as described above, when pure, freshly prepared reagent is needed, they do offer a complementary way of generating the required material. A huge benefit is the fact that standard unmodified polypropylene is used for cartridge fabrication. That means that they can undergo the same decontamination and waste disposal protocols as other widely used PP consumables such as pippette tips, plastic pippettes, well plates and so on.

## Discussion

Here we have expanded the concept of digitization of organic syntheses, and successfully demonstrated its application to prepare reagents which are of high demand in the modern chemistry and biology labs. Different classes of compounds were chosen to demonstrate the versatility of our approach. All developed protocols were proved to be reproducible and consistent, giving compounds with moderate to high yields and purity being adequate for their further application in synthesis. In two cases ($Pd_2dba_3$ and DMP), the use of reactionware cartridges successfully addressed the problems of handling/preparation of unstable compounds along with more efficient distribution of researchers' time. For the diazirine-NHS synthesis the main driving force to use the reactionware would be the huge economic benefit, as the compound is mainly used in biology labs which are usually neither equipped for running organic reactions nor have the qualified specialists to perform the synthesis. Given the high price of this substance if purchased from commercial source and the ease of operation of cartridge device, even the researchers with basic chemistry training and no dedicated organic chemistry equipment (such as Schlenk lines or rotary evaporators) should be able to prepare the compound on bench and use it immediately. Besides that, the diazirine-NHS is unstable to light because of

photoactive N = N moiety. This rules out the option of keeping large amount of it in stock. By using reactionware, one can just keep a stock of pre-loaded sealed cartridges, because the starting material, levulinic acid, is a perfectly stable compound. This would enable one to quickly generate any substance if needed without any unnecessary logistic delays. In future work we will aim to develop a collaborative network of chemical-digitizers who will be trained to develop methodologies to be built into the chemical generators, and also these will be cross-validated by different groups at different locations. By doing this we will ensure that the approach becomes accessible to researchers as soon as possible.

## Methods

**General experimental remarks.** Solvents and reagents were used as received from commercial suppliers unless otherwise stated. Polypropylene feedstock for 3D printing was purchased from Barnes Plastic Welding Equipment Ltd, Blackburn, UK. 3D printing was achieved on Ultimaker 2+ FDM 3D printers supplied by Ultimaker and modified by the authors to print with polypropylene. $^1H$, $^{13}C$ and $^{31}P$ NMR spectra were recorded on a Bruker Avance III HD 600 MHz and Bruker Avance II 400 MHz spectrometers. Mass Spectra were recorded on a Q-trap, time-of-flight MS (MicroTOF-Q MS) instrument equipped with an electrospray (ESI) source supplied by Bruker Daltonics Ltd. All analysis was collected in positive ion mode. The spectrometer was calibrated with the standard tune-mix to give a precision of ca.1.5 ppm in the region of $m/z$ 100–3000. Percentage purity was assessed on a Dionex 3000 Ulitmate HPLC system comprising LPG-3400SD pump, DAD-3000 detector with a 13 μL flow cell, WPS-3000TFC analytical autosampler with fraction collector, and TCC-3000SD column thermostat, running Chromeleon 6.8. A reversed-phase C18 column (Purospher® STAR RP-18 endcapped (5 μm), $100 \times 4.6$ mm) was used. In order to provide the maximum possible information for the reproduction of the work described in this study, a substantially more detailed account of the methods and materials used is provided in the associated Supplementary Methods.

**Traditional glassware syntheses.** The glassware-based synthesis of Compound A, succinimidyl 4,4′-azipentanoate, was achieved by an adapted literature procedure[51] in a three-step process; the Supplementary Methods gives full details of these procedures along with analysis of the final product and intermediate steps. The glassware-based synthesis of Compound B, tris(dibenzylideneacetone)dipalladium(0), was adapted from literature procedures[20,52]. To prevent decomposition of the $Pd_2dba_3$, chloroform used for the synthesis was purified to remove the possible traces of HCl. This was achieved by washing of the chloroform three times with deionized water, followed by pre-drying with magnesium sulfate and finally refluxing over phosphorus pentoxide for 1 h, followed by distillation. The freshly distilled chloroform was stored over well dried 3 Å molecular sieves, in a dark space. The complete synthesis is fully described in Supplementary Methods 1.2, along with relevant analytical data. The glassware-based synthesis of compound C, 1,1,1-triacetoxy-1,1-dihydro-1,2-benziodoxol-3(1H)-one (Dess-Martin Periodinane) was achieved by a two-step procedure adapted from the literature[34] and described in detail in the Supplementary Methods along with all relevant analytical data on the final product and intermediate materials. The glassware synthesis of compound D, the $\{P_8W_{48}\}$ POM cluster, was carried out according to a three-step synthesis adapted from a literature procedure[53]. A detailed account of the synthesis can be found in the Supplementary Methods along with relevant analytical data for the final product and all intermediate steps.

**3D printing of reactionware cartridges.** All reactors used in this study were designed using OpenSCAD (http://www.openscad.org/) based software and printed on Ultimaker 2 +3D printers (https://ultimaker.com/), with 0.6 mm nozzles using polypropylene from a local supplier listed above. The designs for the synthesis cartridges were exported as stereolithography (.stl, provided as Supppplementary Data 1) files and translated into 3D printer instruction files using Cura (https://ultimaker.com/en/products/cura-software), a freely available slicer software

package developed by Ultimaker. These instruction files were then transferred to the 3D printer for fabrication. Devices were printed at 260 °C on 12-mm-thick polypropylene plates with a three-layer raft extending 12 mm outside the model footprint to avoid warping. To allow the introduction of necessary reagents, starting materials, or non-printed components, the printing process was modified to pause at pre-programmed intervals during the fabrication to allow their placement. Once cartridge fabrication was complete the cartridges were flushed with a suitable inert gas (dry $N_2$ supplied by BOC) and sealed prior to use.

**Design and optimization of reactionware cartridges**. Initially, each step of the reaction was carried in a separate module to ensure that the intermediate reaction products met the expected standards. Optimal cartridge volumes for each step of the entire process were also determined at this stage. Once an optimal architecture and volume was found for each step of the entire synthesis, the modules were combined together one by one stepwise to build up the final sequence. Following this approach an entire monolithic system of cartridges for the synthesis of each target compound was created and fabricated. For the Cartridge designed for the synthesis of Compound C, the experimental setup for the glassware synthesis required fitting of a drying tube to the reaction flask. In order to include a similar functionality to the reactionware vessel, a side chamber was added to the upper part of the main chamber. The smaller side chamber was then filled in with a drying agent, such as fused calcium chloride. To stop any drying agent falling into the main reaction chamber, some cotton wool was placed in the channel connecting both chambers.

**Reactionware syntheses**. The syntheses of the target compounds were achieved in the reactionware cartridges by following a set of detailed step-by-step operations at defined intervals. These operations are collected in Operation sequence tables which can be found in Supplementary Methods along with all relevant analytical data for the materials produced by these sequences.

**Product validation experiments**. In order to demonstrate the usability of the chemicals produced by the reactionware cartridges we performed a series of experiments to benchmark the behaviour of the materials obtained against fresh commercial samples of the same materials. In each case the benchmarking experiments were carried out under identical conditions for the reactionware-generated and commercially obtained materials.

NHS-diazirine (6 mg, 0.027 mmol) and alanine (9.5 mg, 0.1 mmol, 4 equiv) were loaded into a glass vial followed by addition of 0.9 mL of PBS buffer made up in $D_2O$. The vial was capped and the mixture was stirred at 40 °C overnight. The reaction mixture was then allowed to cool to room temperature, transferred to a quartz vial and stirred while being exposed to UV light (250 W Oriel instruments Xe-bulb) for 20 min. A sample of the mixture was analysed by $^1H$ NMR before and after irradiation. These experiments revealed formation of the intermediate indicated by a shift of the 1.52 ppm doublet peak belonging to free alanine to 1.42 ppm doublet peak belonging to the diazirine–alanine intermediate (this diazirine–alanine intermediate was also observed by ESI-MS(+ve) $m/z$: calc. for $C_8H_{13}N_3O_3$ 200.21 $[M+H]^+$, found 200.152 $[M+H]^+$). The $^1H$ NMR spectrum of the mixture after UV irradiation showed formation of complex mixture of products, though one crucial observation revealed shift of 2.15 ppm triplet of the proton adjacent to the former diazirine moiety to a new position at 2.41 ppm indicating the activation and subsequent reaction of the diazirine moiety. No differences were observed between experiments conducted with commercially available NHS-diazirine and that synthesized in the reactionware system.

To validate the efficacy of compound B, a Suzuki coupling reaction was carried out according to a published procedure[54]. The experiment carried out with $Pd_2dba_3$ from an external supplier and experiment carried out with reactionware synthetized catalyst both gave exactly the same conversion ratio to the Suzuki product (94%) as determined by $^1H$ NMR.

To validate the efficacy of compound C, an oxidation reaction comparison was carried out as follows: menthol (0.276 g, 1.77 mmol, 1.5 equiv) was loaded into a round bottom flask and followed by 6 mL of DCM. The content of the flask was stirred to achieve complete dissolution of menthol. DMP (0.5 g, 1.18 mmol, 1 equiv) was then added in and the vessel was closed with a rubber septum. The reaction mixture was then stirred for 45 min at room temperature. Sodium sulfite 25% w/v (2 mL) was then added to the reaction vessel, along with 2 mL of a saturated aqueous solution of sodium bicarbonate. The resulting mixture was stirred for 30 min. The reaction mixture was extracted with diethyl ether (25 mL), the organic phase dried over magnesium sulfate, followed by filtration and solvent evaporation. Proton NMR of the crude reaction mixture indicated 77% conversion of menthol to menthone in the experiment carried out with DMP synthetized in reactionware and 82% conversion when carried out with the DMP purchased from an external supplier. Finally, as compound D is not commercially available, validation was performed by repeating a literature synthesis which uses it as a starting material[43]. The synthesis was carried out as described apart from the filtration procedure. The filtration was performed on a hot solution instead of waiting for it to cool down as suggested in the reference. Once the mother liqueur was prepared, after few hours crystals started to appear. The solution was left to crystallize for a day and blue block crystals of sufficient quality were collected and

analysed using an X-ray diffractometer. Unit cell: $a = b = 24.116$, $c = 21.810$; $\alpha = \beta = \gamma = 90°$. Space group I4/M. The unit cell reported in the literature[43] has the axis of $a = b = 26.753$, $c = 21.810$ with $\alpha = \beta = \gamma = 90°$, and the same space group as the unit cell we report –I4/M. There is a small difference in the $a$ and $b$ axes but there is no ambiguity regarding the incorporation of a Cu cluster within the $\{P_8W_{48}\}$ framework as shown by X-ray diffraction structure.

## Data availability
The authors declare that data supporting the findings of this study are available within the paper, its Supplementary Information files and upon request from the authors.

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

## Acknowledgements

The authors gratefully acknowledge financial support from the EPSRC (Grant Nos EP/H024107/1, EP/I033459/1, EP/J00135X/1, EP/J015156/1, EP/K021966/1, EP/K023004/1, EP/K038885/1, EP/L015668/1, EP/L023652/1), and the ERC (project 670467 SMART-POM). Some of this research was developed with funding from the Defence Advanced Research Projects Agency (DARPA) in the 'make it' program. The views, opinions and/or findings expressed are those of the authors and should not be interpreted as representing the official views or policies of the Department of Defence or the U.S. Government DARPA. The authors would like to acknowledge the contribution of Qi Zheng and Ross Winter for their assistence in synthesis of the {W$_{48}$} cluster. Also, Deliang Long for help with XRD data processing and Jim McIver for help setting up the lab for reactionware synthesis.

## Author contributions

L.C. invented the concept, devised the project and the digitization approach. P.J.K. and S.S.Z. did initial development and CAD designs, A.B. and P.F. did CAD design and synthesis for specific targets, S.S.Z. wrote the paper with help from P.J.K., A.B., P.F. and L.C. P.F. and A.B. set up the lab for reactionware synthesis.

## Competing interests

The authors declare no competing interests.
