## [Peer Review File · Nature Communications]

Reviewers' comments:

Reviewer #1 (Remarks to the Author):

This manuscript from Cronin and co-workers describes the use of 3D-printed reactors, called « chemical generators », as substitute of traditional glassware for the synthesis of Pd₂dba₃, diazirine-NHS, DMP, and P8W₄₈ POM. This work is an extension of the previous work published by the same group last year (Science 2018). To my opinion, this work does not deserve publication in NC.

1. The term "chemical generator" is, if not misleading, at least overblown as the cartridges used are no more than chemical reactors.
2. In Figure 1, authors compare the cost of three chemicals obtained either from supplier or by in-house preparation. However, the values are not supported by any rational calculations. Authors should explain how they calculated the prices depicted in Figure 1. They should explain which parameters they included: manpower, energy, depreciation expense, waste treatment, 3D-printing...
3. The preparation of Pd₂dba₃ is confusing as Figure 3 reports the use of Pd(OAc)₂ while the use of PdCl₂ is mentioned in the text.
4. Page 10 "by reacting levulinic acid" or "by reaction of levulinic acid".
5. I'm concerned with the sustainability of such approach as cartridges are disposable and not reused and recyclable. A detailed comment on how this approach competes favorably (or not) with traditional synthesis with regard to sustainability could help the reader in the evaluation of this technology.
6. I'm not convinced by the conclusion which highlights the ease of use of these cartridges. As their use is mainly based on manual operations and not automated with robots, I don't see any groundbreaking simplification (required for operators having basic knowledge of chemistry).
7. Authors justify the use of cartridge as they can be pre-loaded and stocked/stored in the lab. However, the same argument was used as a detrimental comment to qualify "modern synthetic chemistry". The storage of pre-loaded cartridges does not address issues associated with stability and storage organization. Storing chemicals in cartridges or glass bottles does not significantly modify the lab organization.

Reviewer #2 (Remarks to the Author):

Cronin and coworkers demonstrate the use of their 3-D printed reactors for preparation of 4 reagents, three of which are commonly used in organic synthesis. The Authors do a nice job of outlining the benefits of the printed reactors and benchmarking their utility for these reagent preps. I do not believe this application demonstration meets the novelty, urgency, or advance that is necessary for publication in Nature Communications. This would be much more appropriate for a more focused journal such as Organic Letters or European Journal of Organic Chemistry.

Reviewer #3 (Remarks to the Author):

This paper describes the digitization of chemical reagent synthesis with 3D printable cartridges and easy to follow instructions. It is a disruptive advance that has the potential to powerfully enable the democratization of molecule making. The authors correctly make the case that for issues of cost and/or lack of shelf stability, many chemistry labs choose to prepare reagents that are already commercially available, and this technology could help save time and cost in such situations. It is also true that non-chemistry labs are usually limited to only using reagents that are commercially available, and this technology has the potential to expand non-specialist access to useful chemical reagents that are described in the literature but are not commercially available. This, in turn, has the potential to broadly enable research in fields outside of chemistry. There is another area of potential

impact of this technology...as building block-based automated synthesis platforms continue to mature, many of the most versatile building blocks have become commercially available, which has powerfully enabled utilization. There is also, however, an increasing demand for some "specialty building blocks" that are needed, in combination with commercially available blocks, to complete the synthesis of a particular target, and because these specialty blocks are less commonly used, the driving forces for their commercial availability are not as strong. This digitization/cartridge technology described in this article could enable many such building blocks to be digitally encoded for on-demand production as needed, substantially expanding the power of building block-based small molecule construction. The authors show 4 different examples of on-demand reagent synthesis in this report. Notably, the 4 targets are quite diverse, which argues to the versatility of the approach. The characterization in the SI is strong. All this said, there are some limitations as well. These latest generation printed devices require some parts that are not printed and must be purchased, potentially creating some barriers to adoption, but the parts (screws and glass frits) seem to be readily available from multiple vendors. Some of the purities might be suboptimal for utilization, and some of the manipulations required for purification may be a bit laborious and subject to variability. All this said, the authors make a strong case that the procedure performed with the cartridges are more efficient than with standard glassware. I strongly support publication of this manuscript in Nature Communications, after the following recommendations for improvement are considered:

1) The issue about purity is probably the most important and could be readily addressed by demonstrating utility of the reagents that have been prepared using the cartridges. At the end of the day, it's all about function, and rather than quibble about what is "pure enough", the authors could just show that the reagents that were prepared using their cartridges can do what they are supposed to do (ie repeat known applications of these reagents using the material directly as prepared in the cartridges, and compare the results to commercially obtained reagents). This would likely have a big impact on increasing excitement and encouraging adoption of this technology among the broader community.

2) It would be powerful to have another lab demonstrate that the code is sufficient to reproduce the procedure for printing/assembling the cartridge and using it to prepare one of these reagents in usable form, ie like a nextgen/digital version of the classic the Org Syn model.

Minor suggestions:

3) It would be helpful to include pictures and digital cutaways of all 4 of the cartridges in the manuscript - it will help make the technology/concept more accessible to non-specialists, and also emphasize the versatility of the approach.

4. I might have missed it, but the authors should include a link to the digital code for each reagent prep. Can this be included in each of the corresponding Figure legends and/or directly in the text to maximize immediate accessibility?

Reviewers' comments in italics, our replies in normal type.

Reviewer #1 (Remarks to the Author):

This manuscript from Cronin and co-workers describes the use of 3D-printed reactors, called « chemical generators », as substitute of traditional glassware for the synthesis of Pd2dba3, diazirine-NHS, DMP, and P8W48 POM. This work is an extension of the previous work published by the same group last year (Science 2018). To my opinion, this work does not deserve publication in NC.

We disagree. Whilst the previous work showed the proof of concept for the idea producing baclofen, the proof that we could take useful synthesis protocols and turn them into viable chemical generators that would have utility was not proven. In this work proves the generality of our approach and its usefulness to people needing these compounds.

1. The term “chemical generator” is, if not misleading, at least overblown as the cartridges used are no more than chemical reactors.

We would disagree with this judgement, as, unlike the chemical reactors, the cartridges presented often consist of multiple individual reactors with interconnections and have in-built separation/purification capabilities yielding not just the crude chemical substance (like batch reactors do), but rather a ready to use high purity reagent. Thus, they are much closer to miniaturized chemical factories. The term “chemical generator” originates from studies by Kappe et. al. (see, for example, 10.1021/acs.joc.6b01190). It is used here instead of “reactor” exactly to emphasize that the setup yields a pure ready-to-use chemical product. This claim is bolstered by the additional experiments we have included that show the products can be used directly from the chemical generator and perform as expected.

2. In Figure 1, authors compare the cost of three chemicals obtained either from supplier or by in-house preparation. However, the values are not supported by any rational calculations. Authors should explain how they calculated the prices depicted in Figure 1. They should explain which parameters they included: manpower, energy, depreciation expense, waste treatment, 3D-printing...

The paper says that the values given represent “The material (i.e. excluding human resource/infrastructure) cost difference between in house synthesis and buying a reagent from a company”. As such, the prices depicted in the Figure 1 are simple representations of the retail cost of the product per moles as that is representative of the cost to a laboratory to obtain the reagent in its final form ready for use. We have contrasted that with the cost of the starting materials (excluding solvent and other ancillary service costs) necessary to obtain one mole of the target compound. We fully agree that these are not the only costs necessary to obtain the target compounds by in-house synthesis, but further factoring in of ancillary costs such as human resources, electricity and solvents will inevitably be very subjective and depend heavily on the particular laboratory in which the synthesis has been performed. We recognise that this approach could lead to some confusion over the basis of the comparison being made, and so we have amended the manuscript to clarify this issue.

3. *The preparation of Pd₂dba₃ is confusing as Figure 3 reports the use of Pd(OAc)₂ while the use of PdCl₂ is mentioned in the text.*

We thank the reviewer for this note, this typo has been corrected in the manuscript.

4. *Page 10 “by reacting levulinic acid” or “by reaction of levulinic acid”.*

We thank the reviewer for this note, this typo has been corrected in the manuscript.

5. *I’m concerned with the sustainability of such approach as cartridges are disposable and not reused and recyclable. A detailed comment on how this approach competes favorably (or not) with traditional synthesis with regard to sustainability could help the reader in the evaluation of this technology.*

We fully agree with the reviewer that sustainability is an important issue and must be fully addressed. These days lots of laboratory consumables (single-use pipettes, pipette tips, Eppendorf tubes etc.) are being made from polypropylene as it is very cheap, chemically inert and fairly reusable polymer. As long as the reactionware cartridges are made from polypropylene as well, they can completely follow the same decontamination/disposal protocols which are already in place for the rest of polypropylene labware.

Also in the future, the use of chemical generators will expand and with it will the industry that sets about lowering the cost / sustainability. The fact that the system reduces labour costs and enables access beyond conventional labs is also very important.

6. *I’m not convinced by the conclusion which highlights the ease of use of these cartridges. As their use is mainly based on manual operations and not automated with robots, I don’t see any groundbreaking simplification (required for operators having basic knowledge of chemistry).*

The aim is that eventually the system will be usable by non-chemists with minimum of infrastructure – we have highlighted this in the abstract. In this respect the invention of reactionware is a true breakthrough. There are several important points here. First, as we effectively limit all possible operations to only a bounded subset, only limited training is required from the operator to follow the sequence (the detailed sequences of operations are listed in the SI). For example, no knowledge how to do phase separation & no tricks how to make the phases to separate are needed – all this is intrinsically contained within the cartridge. Second, because the person only deals with one operation at a time, chances to make a mistake due to lack of attention are significantly reduced. For example, when adding the reagent you don’t have to watch precipitation, monitor the temperature and maintain the required addition rate simultaneously as you do with traditional glassware.

7. *Authors justify the use of cartridge as they can be pre-loaded and stocked/stored in the lab. However, the same argument was used as a detrimental comment to qualify “modern synthetic chemistry”. The storage of pre-loaded cartridges does not address issues associated with stability and storage organization. Storing chemicals in cartridges or glass bottles does not significantly modify the lab organization.*

We apologise for not being clear, the crucial issue is the time needed for the synthesis. Of course, one can stock all stable precursors for any synthesis supplied by any major company in standard glass bottles. However, this wouldn’t be practical due to multiple reasons as outlined in the paper.

First, many reagents often come in higher minimal volume than might be needed for any particular lab. It's not convenient to buy and store a 250 mL bottle if you only need 50 mL, not mentioning the associated extra costs, waste and so on.

Secondly, to run any synthesis you will have to measure/weigh all the chemicals, prepare the glassware, run manual inter-step purifications and clean the glassware in the end. With the reactionware system these interventions are reduced to pumping the solvent through the system in the right order yielding significant time benefits. As we sum up in the end of the paper, the exact amount of time saved is, of course, rather subjective metric and it depends heavily on one's skills. However, one can't deny that time needed to run the reaction is cartridge is much shorter compared to any glassware protocol.

Third, as rightfully noticed by reviewer #3 below, the benefit of proposed approach is enabling non-specialists (e.g. biologists) to access the reagents that otherwise might not be accessible due to limited budget.

Reviewer #2 (Remarks to the Author):

Cronin and coworkers demonstrate the use of their 3-D printed reactors for preparation of 4 reagents, three of which are commonly used in organic synthesis. The Authors do a nice job of outlining the benefits of the printed reactors and benchmarking their utility for these reagent preps. I do not believe this application demonstration meets the novelty, urgency, or advance that is necessary for publication in Nature Communications. This would be much more appropriate for a more focused journal such as Organic Letters or European Journal of Organic Chemistry.

We thank the reviewer for their comments regarding the benefits, benchmarking and utility of the work presented. We do, however, disagree with regards to the novelty, urgency and advance represented by the current study. The current work represents a significant advance on our previous work in the area with the addition of new design features (inter-chamber valving) and a demonstration of not just the variety of chemical products able to be synthesized (e.g. inorganic cluster materials) but demonstrating the case for wider adoption of the technology being developed. Further, the more focussed journals suggested for publication are focussed more exclusively on the mechanisms of specifically organic synthesis and as such, we believe the work meets the requirements for publication in Nature Communications as we believe it to be of great interest to the wider chemistry and scientific community. It is important that the access to important chemicals on demand is addressed way beyond the specialist literature and this innovation will not only have relevance to chemists, but many people who rely on chemists to produce important compounds for them in use in other fields like molecular biology, materials science, forensics, condensed matter physics, pharmacology, and so on.

Reviewer #3 (Remarks to the Author):

This paper describes the digitization of chemical reagent synthesis with 3D printable cartridges and easy to follow instructions. It is a disruptive advance that has the potential to powerfully enable the democratization of molecule making. The authors correctly make the case that for issues of cost and/or lack of shelf stability, many chemistry labs choose to prepare reagents that are already commercially available, and this technology could help save time and cost in such situations. It is also true that non-chemistry labs are usually limited to only using reagents that are commercially available, and this technology has the potential to expand non-specialist access to useful chemical reagents that are described in the literature but are not commercially available.

This, in turn, has the potential to broadly enable research in fields outside of chemistry. There is another area of potential impact of this technology...as building block-based automated synthesis platforms continue to mature, many of the most versatile building blocks have become commercially available, which has powerfully enabled utilization. There is also, however, an increasing demand for some "specialty building blocks" that are needed, in combination with commercially available blocks, to complete the synthesis of a particular target, and because these specialty blocks are less commonly used, the driving forces for their commercial availability are not as strong. This digitization/cartridge technology described in this article could enable many such building blocks to be digitally encoded for on-demand production as needed, substantially expanding the power of building block-based small molecule construction.

We thank the reviewer for the valuable comment. Indeed, the approach chosen can be used to implement any custom-designed building blocks synthesis. The choice of examples presented in the paper was focussed on demonstrating the versatility of the approach, as noted below.

The authors show 4 different examples of on-demand reagent synthesis in this report. Notably, the 4 targets are quite diverse, which argues to the versatility of the approach. The characterization in the SI is strong. All this said, there are some limitations as well. These latest generation printed devices require some parts that are not printed and must be purchased, potentially creating some barriers to adoption, but the parts (screws and glass frits) seem to be readily available from multiple vendors. Some of the purities might be suboptimal for utilization, and some of the manipulations required for purification may be a bit laborious and subject to variability.

As discussed above we have conducted further experiments now detailed in the manuscript and SI which demonstrate the utility of the generated reagents. Whilst we do agree somewhat with the comment on manipulations, it is important to keep in mind that these manipulations are replacing even more laborious and variable procedures of traditional organic synthesis. As such the new routines represent a significant *relative* improvement in this area. Also we are working towards a version of this design would be ultimately driven by a machine and not by a technician pushing the syringes, which should resolve the concerns about manipulations complexity, and to which the itemised precise instructions of the current system represent both a significant stepping stone, whilst also being a low-cost, low-tech implementation of the system which lowers the barriers to widespread adoption.

All this said, the authors make a strong case that the procedure performed with the cartridges are more efficient than with standard glassware. I strongly support publication of this manuscript in Nature Communications, after the following recommendations for improvement are considered:

1) The issue about purity is probably the most important and could be readily addressed by demonstrating utility of the reagents that have been prepared using the cartridges. At the end of the day, its all about function, and rather than quibble about what is "pure enough", the authors could just show that the reagents that were prepared using their cartridges can do what they are supposed to do (ie repeat known applications of these reagents using the material directly as prepared in the cartridges, and compare the results to commercially obtained reagents). This would likely have a big impact on increasing excitement and encouraging adoption of this technology among the broader community.

We fully agree with this remark. In order to prove that the purity of prepared reagents is enough for their targeted applications we've run a test reaction for each of the reagents prepared and amended the manuscript and SI with the corresponding data.

2) It would be powerful to have another lab demonstrate that the code is sufficient to reproduce the procedure for printing/assembling the cartridge and using it to prepare one of these reagents in usable form, ie like a nextgen/digital version of the classic the Org Syn model.

This is a great idea and whilst we don't feel it is crucial for this work, it is crucial going forward and we are making plans to do this and we indeed have plans for collaborations on this project on the roadmap; we have mentioned this in the revised manuscript. Our end aim is for the cartridges to be driven by a dedicated machine (which is currently being developed) and not by a human. As soon as the cartridge machine comes into an active testing stage, we indeed plan to check the reproducibility of the digital code between operators. In the meantime, we are developing potential new collaborations, and crucially, the transformations to be done in the chemical generators. To highlight this we have added some sentences to the future work at the end of the manuscript.

Minor suggestions:

3) It would be helpful to include pictures and digital cutaways of all 4 of the cartridges in the manuscript - it will help make the technology/concept more accessible to non-specialists, and also emphasize the versatility of the approach.

The architectural details of the reactionware cartridges used for each of the targets presented are available in full in the Supplementary Information. We believe that the inclusion of all of these designs into the main manuscript would merely clutter the presentation of our concept, but we have included such diagrams for the NHS-diazirine cartridge (manuscript figure 4) and partially for the Pd₂dba₃ cartridge (manuscript figure 3). We believe this, combined with the full information in the supplementary Information is sufficient but we are open to further suggestions.

4. I might have missed it, but the authors should include a link to the digital code for each reagent prep. Can this be included in each of the corresponding Figure legends and/or directly in the text to maximize immediate accessibility?

We thank the reviewer for raising this important point. All the operation sequences required to reproduce each of the syntheses are fully presented in the SI for the paper. All the STL files required to create the cartridges are provided as separate SI attachments.

REVIEWERS' COMMENTS:

Reviewer #1 (Remarks to the Author):

This revised version tentatively addresses the remark of the referees. I'm still not convinced by the arguments of the authors for the following reasons (see below). I do not support this paper for publication in NC.

1. Referee. This manuscript from Cronin and co-workers describes the use of 3D-printed reactors, called «chemical generators», as substitute of traditional glassware for the synthesis of Pd2dba3, diazine-NHS, DMP, and P8W48 POM. This work is an extension of the previous work published by the same group last year (Science 2018). To my opinion, this work does not deserve publication in NC.

Authors. We disagree. Whilst the previous work showed the proof of concept for the idea producing baclofen, the proof that we could take useful synthesis protocols and turn them into viable chemical generators that would have utility was not proven. In this work proves the generality of our approach and its usefulness to people needing these compounds.

Referee. I don't see any significant difference between the previous work and the current manuscript. The same type of 3D-printed cartridges are used and I don't understand the argument justifying that the "proof we could take useful synthesis protocols and turn them into viable chemical generators that would have utility was not proven". Through the synthesis of Baclofen the proof was obviously given.

2. Referee. The term "chemical generator" is, if not misleading, at least overblown as the cartridges used are no more than chemical reactors.

Authors. We would disagree with this judgement, as, unlike the chemical reactors, the cartridges presented often consist of multiple individual reactors with interconnections and have in-built separation/purification capabilities yielding not just the crude chemical substance (like batch reactors do), but rather a ready to use high purity reagent. Thus, they are much closer to miniaturized chemical factories. The term "chemical generator" originates from studies by Kappe et. al. (see, for example, 10.1021/acs.joc.6b01190). It is used here instead of "reactor" exactly to emphasize that the setup yields a pure ready-to-use chemical product. This claim is bolstered by the additional experiments we have included that show the products can be used directly from the chemical generator and perform as expected.

Referee. Again I'm not convinced by these arguments. I would add that the cartridge have actually no purification capabilities, or at least no more than traditional batch reactors, as the only possible purification is the separation of crystals from the remaining liquid after sometimes at a low temperature. This is absolutely similar than putting a flask in the fridge for crystal growth and later filter on a standard frit. However, this point is not crucial for the acceptance (or not) of the paper as I consider that it is only a matter of speaking jargon.

3. Referee. In Figure 1, authors compare the cost of three chemicals obtained either from supplier or by in-house preparation. However, the values are not supported by any rational calculations. Authors should explain how they calculated the prices depicted in Figure 1. They should explain which parameters they included: manpower, energy, depreciation expense, waste treatment, 3D printing...

Authors. The paper says that the values given represent "The material (i.e. excluding human resource/infrastructure) cost difference between in house synthesis and buying a reagent from a

company". As such, the prices depicted in the Figure 1 are simple representations of the retail cost of the product per moles as that is representative of the cost to a laboratory to obtain the reagent in its final form ready for use. We have contrasted that with the cost of the starting materials (excluding solvent and other ancillary service costs) necessary to obtain one mole of the target compound. We fully agree that these are not the only costs necessary to obtain the target compounds by in-house synthesis, but further factoring in of ancillary costs such as human resources, electricity and solvents will inevitably be very subjective and depend heavily on the particular laboratory in which the synthesis has been performed. We recognise that this approach could lead to some confusion over the basis of the comparison being made, and so we have amended the manuscript to clarify this issue.

Referee. I very sorry but comparing the cost of in house synthesis, taking only the cost of starting material in consideration, with the cost of the product bought from a company is not relevant. The price of a product bought from a company does not only include the cost of starting material. It obviously includes what we could call "the production cost" (starting material, manpower, energy consumption...) and a profit margin. Authors cannot only use the cost of starting material to compare like with like.

4. Referee. I'm concerned with the sustainability of such approach as cartridges are disposable and not reused and recyclable. A detailed comment on how this approach competes favorably (or not) with traditional synthesis with regard to sustainability could help the reader in the evaluation of this technology.

Authors. We fully agree with the reviewer that sustainability is an important issue and must be fully addressed. These days lots of laboratory consumables (single-use pipettes, pipette tips, Eppendorf tubes etc.) are being made from polypropylene as it is very cheap, chemically inert and fairly reusable polymer. As long as the reactionware cartridges are made from polypropylene as well, they can completely follow the same decontamination/disposal protocols which are already in place for the rest of polypropylene labware. Also in the future, the use of chemical generators will expand and with it will the industry that sets about lowering the cost / sustainability. The fact that the system reduces labour costs and enables access beyond conventional labs is also very important.

Referee. That's true that these days lots of laboratory consumables, and probably too much (single-use pipettes, pipette tips, Eppendorf tubes etc.) which are disposable. However, cartridge should not be compared to pipettes or Eppendorf tubes but rather with glassware. Most organic chemistry laboratories work with glassware which can be reused hundreds of times, limiting the production of waste. I didn't see any comment in the revised version regarding this point.

6. Referee. I'm not convinced by the conclusion which highlights the ease of use of these cartridges. As their use is mainly based on manual operations and not automated with robots, I don't see any groundbreaking simplification (required for operators having basic knowledge of chemistry).

Authors. The aim is that eventually the system will be usable by non-chemists with minimum of infrastructure – we have highlighted this in the abstract. In this respect the invention of reactionware is a true breakthrough. There are several important points here. First, as we effectively limit all possible operations to only a bounded subset, only limited training is required from the operator to follow the sequence (the detailed sequences of operations are listed in the SI). For example, no knowledge how to do phase separation & no tricks how to make the phases to separate are needed – all this is intrinsically contained within the cartridge. Second, because the person only deals with one operation at a time, chances to make a mistake due to lack of attention are significantly reduced. For

example, when adding the reagent you don't have to watch precipitation, monitor the temperature and maintain the required addition rate simultaneously as you do with traditional glassware.

Referee. I'm still not convinced by the argument of simplification. The supporting information does not support this argument as many manual operations are still required. However, I can add to the credit of this manuscript that the experimental procedure are detailed with great care, allowing for a non-specialist to reproduce this work. Actually these procedures could be compared to those we can find in Organic syntheses or Nature Protocols.

7. Referee. Also the authors claim, with this work, an expansion of "the concept of digitization of organic chemistry". There is nothing digital in this work regarding the chemistry itself. I understand that the design of the cartridge are computer-assisted but not the synthesis.

Reviewer #3 (Remarks to the Author):

The authors have satisfactorily addressed all the issues I raised, and I believe the manuscript is even stronger. The addition of demonstrated utility of three of the reagents generated is a especially impactful.

I caught just one typo:

"In order to demonstarte"

I remind strongly supportive of publication of this disruptive advance in Nature Communications.

REVIEWERS' COMMENTS:

Reviewer #1 (Remarks to the Author):

This revised version tentatively addresses the remark of the referees. I'm still not convinced by the arguments of the authors for the following reasons (see below). I do not support this paper for publication in NC.

1.

Referee. This manuscript from Cronin and co-workers describes the use of 3D-printed reactors, called «chemical generators», as substitute of traditional glassware for the synthesis of Pd₂dba₃, diazirine-NHS, DMP, and P8W₄₈ POM. This work is an extension of the previous work published by the same group last year (Science 2018). To my opinion, this work does not deserve publication in NC.

Authors. We disagree. Whilst the previous work showed the proof of concept for the idea producing baclofen, the proof that we could take useful synthesis protocols and turn them into viable chemical generators that would have utility was not proven. In this work proves the generality of our approach and its usefulness to people needing these compounds.

Referee. I don't see any significant difference between the previous work and the current manuscript. The same type of 3D-printed cartridges are used and I don't understand the argument justifying that the "proof we could take useful synthesis protocols and turn them into viable chemical generators that would have utility was not proven". Through the synthesis of Baclofen the proof was obviously given.

We are sorry for the confusion. The aim of the previous work was to demonstrate the feasibility of running multi-step syntheses in 3D printed cartridges. Though the idea seems obviously simple, it's far more involved than it looks because many factors both from the 3D-printing side and chemistry side have to be considered. This paper established a workflow from a common multi-step organic synthesis protocol to the physical implementation of the process in 3D printed cartridge. That was demonstrated by the synthesis of Baclofen, however whether this technology may be useful for general organic chemists was an open question. In the current paper the aim was to demonstrate the practical applicability of the developed approach along with improved cartridge design. We've chosen the compounds that are widely used in research laboratories on the day-by-day basis and successfully demonstrated that an average-trained person can use this technology with high degree of reproducibility and significant time savings due to the factors discussed in the manuscript.

2.

Referee. The term "chemical generator" is, if not misleading, at least overblown as the cartridges used are no more than chemical reactors.

Authors. We would disagree with this judgement, as, unlike the chemical reactors, the cartridges presented often consist of multiple individual reactors with interconnections and have in-built separation/purification capabilities yielding not just the crude chemical substance (like batch reactors do), but rather a ready to use high purity reagent. Thus, they are much closer to miniaturized chemical factories. The term "chemical generator" originates from studies by Kappe et. al. (see, for example, 10.1021/acs.joc.6b01190). It is used here instead of "reactor" exactly to emphasize that the setup yields a pure ready-to-use chemical product. This claim is bolstered by the additional experiments we have included that show the products can be used directly from the chemical generator and perform as expected.

Referee. Again I'm not convinced by these arguments. I would add that the cartridge have actually no purification capabilities, or at least no more than traditional batch reactors, as the only possible purification is the separation of crystals from the remaining liquid after sometimes at a low temperature. This is absolutely similar than putting a flask in the fridge for crystal growth and later filter on a standard frit. However, this point is not crucial for the acceptance (or not) of the paper as I consider that it is only a matter of speaking jargon.

We are not quite sure what the referee means by "traditional batch reactors", but the paper describes the following purifying procedures been developed and tested:

- *Crystallization*
- *Evaporation*
- *Filtration*
- *Phase extraction*
- *Drying with a drying agent*

It is worth to note that all these steps are undertaken in a single vessel without process interruption. To the best of our knowledge we can't see how all of that can be achieved by putting a flask in the fridge.

3.

Referee. In Figure 1, authors compare the cost of three chemicals obtained either from supplier or by in-house preparation. However, the values are not supported by any rational calculations. Authors should explain how they calculated the prices depicted in Figure 1. They should explain which parameters they included: manpower, energy, depreciation expense, waste treatment, 3D printing...

Authors. The paper says that the values given represent "The material (i.e. excluding human resource/infrastructure) cost difference between in house synthesis and buying a reagent from a company". As such, the prices depicted in the Figure 1 are simple representations of the retail cost of the product per moles as that is representative of the cost to a laboratory to obtain the reagent in its final form ready for use. We have contrasted that with the cost of the starting materials (excluding solvent and other ancillary service costs) necessary to obtain one mole of the target compound. We fully agree that these are not the only costs necessary to obtain the target compounds by in-house synthesis, but further factoring in of ancillary costs such as human resources, electricity and solvents will inevitably be very subjective and depend heavily on the particular laboratory in which the synthesis has been performed. We recognise that this approach could lead to some confusion over the basis of the comparison being made, and so we have amended the manuscript to clarify this issue.

Referee. I very sorry but comparing the cost of in house synthesis, taking only the cost of starting material in consideration, with the cost of the product bought from a company is not relevant. The price of a product bought from a company does not only include the cost of starting material. It obviously includes what we could call "the production cost" (starting material, manpower, energy consumption...) and a profit margin. Authors cannot only use the cost of starting material to compare like with like.

That's exactly the point we tried to address in the discussion of Figure 1 in the manuscript text. Unfortunately, trying to include these factors in the calculation would give even less reasonable figures because the production costs depend on multitude of factors including, for example, geographical region of manufacture.

4.

Referee. I'm concerned with the sustainability of such approach as cartridges are disposable and not reused and recyclable. A detailed comment on how this approach competes favorably (or not) with

traditional synthesis with regard to sustainability could help the reader in the evaluation of this technology.

Authors. We fully agree with the reviewer that sustainability is an important issue and must be fully addressed. These days lots of laboratory consumables (single-use pipettes, pipette tips, Eppendorf tubes etc.) are being made from polypropylene as it is very cheap, chemically inert and fairly reusable polymer. As long as the reactionware cartridges are made from polypropylene as well, they can completely follow the same decontamination/disposal protocols which are already in place for the rest of polypropylene labware. Also in the future, the use of chemical generators will expand and with it will the industry that sets about lowering the cost / sustainability. The fact that the system reduces labour costs and enables access beyond conventional labs is also very important.

Referee. That's true that these days lots of laboratory consumables, and probably too much (single-use pipettes, pipette tips, Eppendorf tubes etc.) which are disposable. However, cartridge should not be compared to pipettes or Eppendorf tubes but rather with glassware. Most organic chemistry laboratories work with glassware which can be reused hundreds of times, limiting the production of waste. I didn't see any comment in the revised version regarding this point.

We fully agree with that and we do not propose the use of polypropylene cartridge as a complete replacement of glassware. The same way as the use of Eppendorf pipette tips do not rule out the use of glass pipettes, we see the cartridges as a useful amendment to an existing glassware kit, especially for reactions sensitive to impurities. There have been quite a few papers up to date, demonstrating that glassware can not always be completely cleaned and the residual amounts of impurities absorbed on the glass walls can dramatically change the course of reaction. In such case, using a new cartridge that would be disposed after run seems a reasonable economic alternative. In order to make our point more clearly we have added this discussion into the revised manuscript (page 17)

6.

Referee. I'm not convinced by the conclusion which highlights the ease of use of these cartridges. As their use is mainly based on manual operations and not automated with robots, I don't see any groundbreaking simplification (required for operators having basic knowledge of chemistry).

Authors. The aim is that eventually the system will be usable by non-chemists with minimum of infrastructure – we have highlighted this in the abstract. In this respect the invention of reactionware is a true breakthrough. There are several important points here. First, as we effectively limit all possible operations to only a bounded subset, only limited training is required from the operator to follow the sequence (the detailed sequences of operations are listed in the SI). For example, no knowledge how to do phase separation & no tricks how to make the phases to separate are needed – all this is intrinsically contained within the cartridge. Second, because the person only deals with one operation at a time, chances to make a mistake due to lack of attention are significantly reduced. For example, when adding the reagent you don't have to watch precipitation, monitor the temperature and maintain the required addition rate simultaneously as you do with traditional glassware.

Referee. I'm still not convinced by the argument of simplification. The supporting information does not support this argument as many manual operations are still required. However, I can add to the credit of this manuscript that the experimental procedure are detailed with great care, allowing for a non-specialist to reproduce this work. Actually these procedures could be compared to those we can find in Organic syntheses or Nature Protocols.

We thank the reviewer for their evaluation of our work. The crucial point here is not the number of operations, but the number of operation types (open valve, close valve, push syringe, wait). Because there are only a few different types of operations, the person doing the synthesis manually needs to have only a

very limited training. In the future, a machine can be easily programmed with this minimal set of instructions and perform them millions of times in various synthetic sequences.

7.

Referee. Also the authors claim, with this work, an expansion of “the concept of digitization of organic chemistry”. There is nothing digital in this work regarding the chemistry itself. I understand that the design of the cartridge are computer-assisted but not the synthesis.

The key concept of the chemistry digitization is presenting the multi-step organic synthesis in a universal digital form. This is a very hard task and many research groups are pursuing it in different ways. The way presented in this paper enables to translate the protocol (sequence of operations) into digital form by reducing it to a set of distinct repeatable instructions. This code for the synthetic sequence along with the digital code for the cartridge geometry itself encapsulate all information necessary to replicate all the chemistry from scratch (without any special prerequisites like glassware) at any time. Thus we think it rightfully demonstrates one of the approaches to the digitization of chemistry.

Reviewer #3 (Remarks to the Author):

The authors have satisfactorily addressed all the issues I raised, and I believe the manuscript is even stronger. The addition of demonstrated utility of three of the reagents generated is a especially impactful.

I caught just one typo: "In order to demonstrarte". I remind strongly supportive of publication of this disruptive advance in Nature Communications.

We thank the reviewer for bringing this up, the typo has been corrected.